# Tn5 DNA Transposase in Multi-Omics Research

**DOI:** 10.3390/mps6020024

**Published:** 2023-02-28

**Authors:** Dmitry Penkov, Ekaterina Zubkova, Yelena Parfyonova

**Affiliations:** 1IRCCS San Raffaele Hospital, 20132 Milan, Italy; 2National Medical Research Centre of Cardiology Named after E. I. Chazov, 121552 Moscow, Russia; 3Faculty of Medicine, Lomonosov Moscow State University, 119991 Moscow, Russia

**Keywords:** Tn5 transposase, omics, tagmentation, CUT&Tag

## Abstract

Tn5 transposase use in biotechnology has substantially advanced the sequencing applications of genome-wide analysis of cells. This is mainly due to the ability of Tn5 transposase to efficiently transpose DNA essentially randomly into any target DNA without the aid of other factors. This concise review is focused on the advances in Tn5 applications in multi-omics technologies, genome-wide profiling, and Tn5 hybrid molecule creation. The possibilities of other transposase uses are also discussed.

## 1. An Overview of DNA Transposases

Transposable elements, or transposons, are nucleic acid sequences that can change their position within a genome. They are the most abundant protein coding sequences in nature and can be found in essentially all prokaryotic and all eukaryotic genomes, as indicated by currently available sequence data. Transposons were discovered in the 1940s by Barbara McClintock, who was studying genetic mutations in maize [1,2]. Transposable elements make up a substantial fraction of genomes and are responsible for variations in the genome size among species. Although transposons are harmful to the host cell in most cases, they can be also useful, donating coding sequences or regulatory elements that are implicated in host gene regulation. It is estimated that roughly a quarter of human promoter regions contain sequences derived from transposable elements [3]. Similar “co-option” can be seen, for example, in the domestication of transposon-containing recombination genes during the evolution of adaptive immunity in vertebrates [4]. Transposons can be divided between (1) retrotransposons, which use RNA as an intermediate molecule in their transposition-based life cycle, and (2) DNA transposons, which solely use DNA intermediates. Transposable elements rely on host cellular machinery for their transcription, but most of them contain genes that encode proteins necessary for their transposition. These proteins are called transposases.

All DNA transposons can be classified into four main classes: DD (E/D) transposons, Y1- and Y2-transposons, serine transposons, and tyrosine transposons [5]. All known DNA “cut-and-paste” transposons (the transposons that are most widely used in biomedical research) are of the DD (E/D) type. The name refers to three acidic amino acid residues important for the catalysis [6]. As the “cut-and-paste” name suggests, they change their genomic locations by being cut out from one chromosomal site and then by being inserted into another one with the action of a corresponding catalytic DNA transposase. Typically, they produce a double strand break by physically releasing the transposon from the donor DNA, and then integrating it into a new location in the acceptor DNA. However, not all DD (E/D) transposases produce DNA double-strand breaks. For example, during the lytic phase in its life cycle, bacteriophage MuA transposase initially cleaves only the 3′-ends of the transposon and then follows the pathway of replicative transposition [7]. Transposases catalyze reactions only at the ends of their own transposons that contain certain DNA sequences, into which the transposase initially binds site-specifically, synapsing the ends, and thus forming an active DNA transposition complex, a transpososome. Most transposases select target sites relatively randomly with only a slight bias towards a certain preferred “consensus” sequence. In some cases, additional proteins may be used in the targeting process, significantly affecting the target site choice [8,9]. DNA cleavage at the transposon ends results in one strand containing a free 3′-OH group (which is called a “transferred strand” as this 3′-OH group plays a crucial role in the subsequent DNA integration reaction) and another strand, which is called a “nontransferred strand”, as it does not become connected to the acceptor DNA [5]. Transposases catalyze DNA cleavage and strand transfer reactions by the use of “in trans” configuration, inferring that the two ends of a transposon are brought in close proximity by multimerization [10]. This is true for all DNA transposases, i.e., the active form of a DNA transposase is a protein−DNA complex, a transpososome, in which the transposase protein is at least in a dimeric but in some cases in a multimeric form. 

The DNA transposases are divided between prokaryotic (such as Tn3, Tn5, Tn7, Tn10, phage Mu, etc., transposases) and eukaryotic (such as Sleeping Beauty, PiggyBac, Hermes, etc., transposases). Most of them, especially eukaryotic ones, possess no or very low transpositional activity in their native form, and many efforts have been made to revive them or to make them more active [11,12]. Phage Mu transposase (MuA), however, being a transposase of a bacteriophage, is highly active already in its wild-type form, as there is no selective pressure to reduce its activity [13]. In fact, when the phage destroys its host, the transposon escapes the cell, thus evading cellular defense mechanisms that would reduce its activity. Yet, even this protein could be substantially enhanced for its transpositional capabilities [13]. The production of many active transposase forms has made it possible to use them in biotechnological applications, for example, as a tool to introduce various genes into a genome [14,15,16,17]. This approach often requires a two-vector system: one plasmid that encodes a transposon bearing a gene to be transferred and another plasmid that encodes the transposase under the control of a suitable promoter (Figure 1). Alternatively, after delivery of a transposon via a plasmid into a cell, the transposase can be delivered as a protein. 

Another way to introduce a transposon and a transposase into a cell is via delivery of in vitro pre-assembled DNA transposition complexes, transpososomes. They are typically delivered into cells by electroporation. This can be done efficiently not only in various bacteria [18,19,20,21], but also in yeast and mammalian cells [22]. Other ways to deliver the plasmids or proteins into cells have also been used, such as transfection, injection, or packaging them within viruses. Many transposases require a multimeric complex, also containing specific host factors, for their action, which makes it difficult or impossible to use them as efficient tools. However, others, such as Tn5, MuA, PiggyBac, and Sleeping beauty, are able to transpose without the need for other auxiliary proteins. In vitro transposition systems, especially those based on Tn5 or MuA transposases, have provided a plethora of strategies for biotechnological applications, ranging from mutagenesis and DNA sequencing applications to protein and genome engineering approaches [13,19,23,24,25,26,27]. In this review, we will focus primarily on Tn5 transposase usage in DNA tagmentation. 

## 2. Tn5-Assisted Tagmentation

The first transposase that was used in an in vitro DNA transposition system was the MuA transposase of phage Mu [28]. These early experiments by Kiyoshi Mizuuchi were performed using plasmid substrates and several cofactors. Later, Mizuuchi and colleagues showed that a much simpler reaction was also possible. That is, two short DNA segments containing transposon-end sequences could be transposed into target DNA in vitro [29]. This process was catalyzed by MuA transposase only, and no other proteins were needed. The fact that MuA transposase does not produce double strand breaks in the donor DNA was circumvented in the strategy by the use of precut transposon ends. Similar approaches using short transposon end sequences for in vitro reactions were soon adopted for many other transposon systems, including Tn5, and this adoption paved the way for the development of a number of biotechnological applications and research approaches. 

An assay for transposase-accessible chromatin using sequencing (ATAC-seq) was invented in 2013 in the Greenleaf laboratory and published in the paper by Buenrostro et al. [30,31]. It is based on the previously described method of Tn5 transposase-assisted tagmentation of DNA, described in [32] and commercialized by Epicentre Biotechnologies Inc. as Nextera library preparation. A similar approach was also developed for Mu-assisted DNA tagmentation and the corresponding reagents are commercially available from Thermofisher Scientific Inc. or Domus Biotechnologies, Finland; the latter company providing enhanced versions of MuA transposase [33].

Tn5 is a bacterial transposase working by a “cut-and-paste” mechanism. The utility of the Tn5 transposon stems in part from its high activity in many different species and lack of target specificity, so that it integrates essentially randomly. It does not require host factors for its activity.

Tn5 possesses an ability to transfer a fragment of DNA flanked by two mosaic ends (ME; 5′-CTGTCTCTTATACACATCT-3′) to a new location, inserting it into double-stranded DNA. The wild-type Tn5 transposase has a relatively low activity and in order to use in biotechnology it was modified to become more active [24,34]. However, most importantly, it was shown that a dimer of the Tn5 transposase protein can be loaded with two double-stranded DNA oligos, each containing the 19 bp sequence of ME, and form a stable complex with them, called a “transpososome”. These 19 bps-long adapters are protected from cutting by the transposase itself [24], but can be extended with a single-stranded overhang, as Tn5 transposase does not cut ssDNA. It was shown that when this complex is added to double stranded-DNA in the presence of Mg^2+^ ions, the DNA is cut in multiple sites and the adaptors are attached in such a way that ssDNA overhang is located on the transfer strand (the strand that becomes covalently bound to the target DNA) adding a “tag” there. The other strand contains 9-nucleotide-long gaps formed due to nicking of DNA before transfer; these gaps can be repaired by a DNA polymerase during a PCR reaction provided that the adapter was phosphorylated at the 5′-end (Figure 2). The transposase complex remains tightly attached to DNA after cutting and needs to be removed before subsequent amplification by PCR. This process of DNA cutting and addition of “tags” was called “DNA tagmentation” and became a milestone in single cell sequencing technologies.

This well-established method, together with the observation that the DNA tagmentation by Tn5 transpososome occurs also in intact nuclei in nucleosome-free regions, inspired the development of ATAC-seq. Originally, the experiments were set up only for bulk cells similar to other sequencing methods. With the development of single cell methods, both scRNA-seq and scATAC-seq appeared. In order to treat all cells together instead of sorting them into separate wells, each cell must be barcoded before the treatment and the most obvious way to do this was to use a pool of Tn5 transpososomes bearing different adaptor overhangs. This is a straightforward procedure in scATAC-seq, where each pair of adaptors has a unique barcode in one of them, but in the case of scRNA-seq has different variations. Here, we will look at these scRNA-seq methods more closely.

## 3. Tn5 Tagmentation in Single Cell Omics Technologies

Initially, even in scRNA-seq experiments, efforts have been made towards full-size mRNA sequencing and therefore limiting the number of total cells processed in a single experiment [35]. In order to make sure that only full-length molecules of double-stranded cDNA were produced, the terminal transferase activity of MMLV-derived reverse transcriptase was exploited. It had been noticed that a stretch of 3”C” nucleotides was added by the reverse transcriptase only when the synthesis of the first strand of cDNA was completed [36,37]. Based on this observation made by the Sandberg group, a so-called “template switching oligo (TSO)” was invented and used as a primer for PCR reactions to amplify full-length cDNA. This method was commercialized by CLONTECH Inc. as Smart-seq (switch mechanism at the 5′-end of RNA templates) technology in their SMARTer Ultra Low RNA kit. Later, this method was improved, keeping the same strategy, and named “Smart-seq2”, and is still frequently used when there is a need to obtain a full-length cDNA prior to sequencing [23]. 

ScRNA-seq methods that instead sequenced only a 3′-end of cDNAs were also developed: Cel-seq and MARS-seq (a modified version of the former) [38,39,40,41,42]. These methods introduce both a barcode (containing a cell and/or molecular identifier) and the T7 promoter at the initial stage and use in vitro transcription (IVT) from the T7 promoter in place of the first PCR amplification. This approach allowed the authors to pool all barcoded cDNA together prior to IVT, and then to fragment RNAs and ligate Illumina adaptors prior to the final PCR. This results in a strong bias to 3′-end fragment enrichment, but increases throughput and reduces cost at the same time.

Another method, called single-cell tagged reverse transcription sequencing (STRT-seq), developed in the laboratory of Sten Linnarsson, used a similar strategy as Smart-seq2, but with some modifications of both oligo(dT) and TSO primers. Both primers contained additional features, including 5′-end biotinylation, that allowed to select both 5′-end and 3′-end of cDNAs, even if only the 5′-end portions were kept and amplified for the sequencing [43,44]. When used for single cell RNA-seq, both Smart-seq2 and STRT-seq methods attach a barcode with cell identifiers after the cDNA amplification step, which implies that cells must be sorted out into individual compartments before the reverse transcription. 

In order to simplify the process of single cell preparation, various methods have been developed to distribute cells in picoliter wells (CytoSeq) or emulsion droplets (Drop-seq and InDrops) [45,46]. In these methods, the lysis buffer was partitioned either by droplets or by array, so that each of the drops contains one bead with a large number of oligos (dT), each of them comprising the same cell identifier, but different molecular identifiers, and no more than one cell (in practice only around 10% of drops contain cells and the other 90% are empty). This approach allowed to process thousands of cells simultaneously, and was commercialized by 10× Genomics as a “chromium platform” [47]. This droplet-based system became the method of choice for thousands of publications focusing on single cell transcriptomic characterization of heterogeneous cell populations with previously sequenced genomes, as the short 3′-end fragments that are obtained do not allow de novo assembly of full-length RNA sequences from single cells. 

In order to fully describe gene regulation at single cell level, transcriptomics characterization itself is not enough. It must be accompanied by other single-cell-based techniques such as ATAC-seq, single cell proteomics, and so on. Indeed, in the last 5 years, numerous methods of multiple omics have been developed [48]. In many of them, Tn5 tagmentation was used as well. Among the first of them, G&T-seq, Target-seq, Sidr-seq, and DR-seq, which address transcriptomic and genomic DNA analysis, genomic libraries were prepared using Tn5 tagmentation, while Smart-seq2 and IVT approaches were used for mRNA analysis [49,50,51,52,53] (Figure 3).

The same is true for scMT-seq and scM&T-seq, two early methods for the simultaneous study of transcriptome and methylome of cells, where Tn5 tagmentation was used only in downstream applications to prepare libraries for sequencing [54,55]. 

There is a large number of methods for simultaneous measurement of open chromatin and transcriptome in single cells. In one of such methods, scCat-seq, cells are first distributed between different wells in a plate, lysed, and the cytoplasmic fraction is taken to another plate keeping the same order of cells [56]. The two fractions then are processed separately. mRNA is converted to cDNA using a Smart-seq2 approach and then tagmented individually for each cell introducing different barcodes. The chromatin was tagmented separately from cDNA using Tn5 transposase in order to access open (“accessible”) regions. This approach, in addition to open chromatin, allows the sequencing of full-length cDNA, but not with a high throughput. 

Methods that increase throughput were developed based on the observation that Tn5 transposase remains tightly bound to DNA in the nucleus after the tagmentation reaction and therefore nuclei can be used as natural compartments for single cell reactions. The mRNA that is located inside a nucleus can be used for transcriptomic analysis, even if a part of the mRNA is missing together with the cytoplasmic fraction. This gave rise to a number of methods studying single nuclei instead of single cells, gaining high throughput. One of the first of them was sci-Car, in which nuclei were first tagmented in bulk using a library of adaptors complexed with Tn5 and then redistributed to different wells for cDNA amplification. Thanks to combinatorial indexing usage, a high-throughput was achieved; as much as 10,000 nuclei in each experiment [57]. 

A similar approach was realized in the Paired-seq method, in which multiple rounds of pooling and splitting followed by barcode ligation were performed, resulting in an ultra-high throughput, making it possible to process up to millions of cells in the same experiment [58]. In both of these methods however, only the 3′-end of cDNA is sequenced for each cell. Similar approaches based on tagmentation of single nuclei and subsequent pooling and splitting were elaborated in several other methods, such as Share-seq, Scito-seq, and Tea-seq [59,60,61]. In Scito-seq, an interesting approach is applied that uses a splint oligo capturing the 3′-end of different oligos attached to different Abs in the same droplet, making it possible to distribute more than one cell per droplet [61]. A combined study of methylome and chromatin accessibility was developed in such methods as scCharm-seq and scNMTseq [62,63,64].

An elegant approach was described in a droplet-based Snare-seq method, where open chromatin tagmentation was performed using a splint oligo attached to a Tn5 adaptor [65]. This oligo contains a poly(A) tail that after tagmentation is captured by an oligo(dT) primer in the same way as mRNA. The oligo(dT) primer contains a cell identification barcode that is identical to both cDNA and gDNA in the same droplet and, therefore, the same cell.

Tn5 tagmentation is also utilized in Astar-seq, a Fluidigm-based method in which single cells are distributed between compartments in microfluidics and then tagmented separately [66]. In the next step, Tn5 transposase is inactivated by EDTA and reverse transcription is performed in the presence of Mg^2+^ ions. Next, cDNA is biotinylated and amplified. Streptavidin beads are used to separate cDNA from the tagmented genomic DNA. In the last step, preparation of both cDNA and gDNA libraries in a standard way is performed.

Tn5 tagmentation is used in Asap-seq, a method that allows simultaneous measurements of chromatin accessibility and surface and intracellular proteins using oligo-labeled antibodies [67]. This method takes advantage of existing antibody reagents used for CITE-seq [68] and introduces bridge oligos that make it possible to process the Abs together with chromatin in the single scATAC-seq procedure.

A significant step forward was recently developed in the Issaac-seq method, which is based on the observation that Tn5 transposase is able to tagment a hybrid DNA-RNA molecule in the same way as dsDNA [69]. A number of earlier studies, described in the Sherry method [70], optimized this process that allowed to fulfill two independent tagmentations: the first one to tagment open regions of chromatin and the second one to tagment RNA/DNA duplexes after reverse transcription, where heterodimers of RNA and DNA are formed. Two separate libraries are then prepared and sequencing is performed.

Therefore, the Tn5 transposase reveals many unique characteristics, such as the ability to form heterocomplexes with any adaptors containing MEs, a tight attachment to DNA after cleavage, and tagmentation of DNA-RNA heterodimers, that have allowed researchers to use it widely in single cell sequencing applications.

## 4. Tn5 Tagmentation in Genome-Wide Profiling Assays

Another area of Tn5 tagmentation applications is genome-wide profiling of protein binding or genomic marks. Historically, one of the most extensively used methods for the study of DNA–protein binding in vivo was ChIP-seq, which gave an enormous amount of information about protein–chromatin interactions and histone modification in vivo [71,72]. However, a major limitation of this method was the requirement of a relatively high cell number; at least a million cells are required for a ChIP-seq analysis. In many cases, to obtain such number of cells is a problematic task. Tn5 tagmentation helped to solve this problem. Contrary to ChIP, in transposase reactions PCR primers are inserted directly into the genomic DNA very close to the binding sites, giving the profile better resolution and more sensitivity. As a result, a genome-wide profiling experiment requires a much lower sequencing depth and number of cells.

The first method in which Tn5 was used to study genome-wide binding properties was TAM-ChIP, a system developed and patented by the Active Motif. In that early method, the chromatin was crosslinked and fragmented by sonication the same way as in ChIP. The TAM-ChIP protocol uses a secondary antibody that is pre-coupled to the Tn5 transposase (Figure 4A). After chromatin capture by agarose beads, Tn5 is activated by Mg^2+^ and PCR primers and transposed to the vicinity of protein binding sites. The important points are that all the binding reactions before the tagmentation must be carried out in the absence of Mg^2+^ to keep the transposase inactive and it must be coupled to the secondary Ab, not the primary one, in order to multiply the number of insertions of the adaptors near the protein binding site. TAM-ChIP allowed to perform genomic targeting and genomic library preparation simultaneously.

The TAM-ChIP protocol was improved, giving rise to the CUT&Tag method [73]. In this approach, the protein-A is directly conjugated with Tn5 transpososome, and this complex is added to not fixed but permeabilized cells, previously sequentially incubated with primary and secondary Abs (Figure 4B). After the protein-A–Tn5 complex binds to a secondary Ab, the transposase is activated by the addition of Mg^2+^ ions, leading to the adaptor insertion into an open chromatin between nucleosomes in the vicinity of the protein binding site. In order to keep the Tn5 inactive in the appropriate buffers until the last stage, the cells are immobilized on concanavalin magnetic beads, making it easy to change buffers between different stages. A great achievement of CUT&Tag was the fact that a native chromatin was used and no sonication and no adapter ligation were required. This led to the extremely low number of cells required for an experiment. The transposase-based approach even made it possible to use it in single cells. Indeed, CUT&Tag was adjusted to single-cell usage as the entire reaction is carried out within intact nuclei [73,74,75,76,77,78]. Instead of using magnetic beads however, the cells were centrifuged between washing steps, and after the tagmentation stage, cells were distributed into different microwells for barcoding.

An interesting and useful application of CUT&Tag is CUTAC (cleavage under targeted accessible chromatin) [79,80]. It takes advantage of the fact that at low salt concentrations, Tn5 has the ability to nonspecifically tagment open chromatin. Thus, by lowering the salt concentration, one can obtain DNA fragments containing both target-specific ends and nucleosome-free nonspecific ends. In the original publication, only two Abs, specific either to Pol2 Serine-5 phosphate or to Polycomb domains (H3K27me3) were used [81]. Under low salt conditions, in both cases, fragments of various sizes were obtained. However, after short fragments were selected in the case of Pol2 Ab, they profiled accessible enhancers and promoters. At the same time, when large fragments were selected in the case of H3K27me3 Ab, silenced regulatory elements were profiled. 

Later, other Abs were used in CUTAC and Abs to transcription factors were also used, but the specific tagmentation events may be low compared to nonspecific ones, as TF binding sites are rare in the genome compared to histone modifications.

CUT&Tag also allowed to enhance the sensitivity of genome-wide profiling methods. While scRNA-seq takes advantage of abundant transcripts of highly and moderately expressed genes, epigenomic profiling is limited to at most two copies of a chromatin feature in each cell. Using the CUT&Tag approach, T7 promotors can be inserted via tagmentation in front of a genomic feature to be profiled and then linear amplified by in vitro transcription. This allows to obtain at least 10 times more unique reads per cell compared with other methods. It was proven to work very well with antibodies to histone marks, PolII, and CTCF transcription factors [82].

A very interesting application of CUT&Tag is the study of the 3D structure of nuclear chromatin. Standard approaches such as ChIA-PET, HiChIP, and PlacSeq require a large number of cells and sequencing reads. The use of CUT&Tag instead of immunoprecipitation in the last step resulted in a 100-fold reduction in the number of cells and a 10-fold reduction in the sequencing depth. The method, called HiCuT, was successfully used to study the CTCF-chromatin structure [83].

A significant drawback of CUT&Tag was always considered to be the inability to use multiple Abs in a single reaction due to the identical protein-A-Tn5 complex in the last step. However, recently this problem was successfully solved in multi-CUT&Tag. In this version of the method, protein-A-Tn5 transpososome carries and deploys adapters with barcodes that are unique to different antibodies [84]. Such an approach allowed different antibodies to be used in the same cells, enabling simultaneous mapping of multiple chromatin-associated protein or histone modifications. Transpososomes containing different barcodes were pre-coupled with corresponding Abs and purified from the excess of the antibodies using TALON beads (as protein-A contains 6-His tag). After this preliminary coupling step, different (up to three) Ab complexes were added to the cells and successfully used in a CUT&Tag reaction. 

This approach, however, showed some cross-enrichments between targets. This drawback was corrected in another elegant approach, in which a primary Ab, instead of being attached to the protein A-Tn5 moiety, was covalently bound to the MEDS adaptors and complexed with pA-Tn5. In such an approach, the adaptor-Ab hybrid molecule could bind to a target directly and then be inserted into the genomic DNA after Tn5 tagmentation [85]. Due to the low number of pA-Tn5 complexes accumulating per target locus in the absence of a secondary antibody, the original method was not very efficient, but later it was modified by addition of a secondary Ab conjugated with the reverse MEDS adaptors complexed with pA-Tn5. This improvement led to a method called MulTI-Tag, that showed very high specificity and efficiency and proved to be applicable to several targets simultaneously and to single-cell CUT&Tag analysis (Figure 5). 

## 5. Tn5-Hybrid Tagmentation

The idea of using hybrid molecules for genomic targeting has been exploited in several methods. One of the first was CUT&RUN, which used a hybrid molecule between protein-A and micrococcal nuclease (MNase) [86,87]. In the CUT&RUN protocol, MNase is fused to protein-A (pA–MNase) to guide the chromatin cleavage to antibodies bound to genomic target sites. CUT&RUN uses isolated nuclei from live (not fixed) cells that are immobilized on lectin-coated magnetic beads. The nuclei are incubated with an antibody specific to the target epitope, followed by the pA–MNase hybrid. The enzymatic reaction of the nuclease is activated by the addition of Ca^2+^ ions. The protein–DNA complex can be isolated and purified and the obtained DNA fragments can be used directly for library preparation.

One very important application of the CUT&RUN protocol is a system to discriminate transcription factor binding to nucleosome-rich and nucleosome-poor chromatin regions that is important in studying the role of transcription factors as pioneer factors. Indeed, when a factor was bound to a nucleosome, the fragments after MNase treatment were larger than the size of the nucleosome, while in the case of binding to an open chromatin, the fragments were of sub-nucleosomic size [88].

Another very successful method was GET-seq, which used a hybrid molecule between Tn5 transposase and HP1-alpha chromodomain [89]. It was shown by the authors that this hybrid was specific to heterochromatic regions of chromatin (in the same way as HP1-alpha) and was able to efficiently tagment the heterochromatin that afterwards can be amplified by PCR. The highly efficient amplification is due to multiple tagmentation sites close to each other, as HP1-alpha binds to H3K9Me3 histone mark, which is abundant in heterochromatic regions. However, the HP1-alpha-Tn5 hybrid also possesses an intrinsic Tn5 activity towards open and inter-nucleosomic regions in euchromatin; therefore, the authors additionally had to use Tn5 alone in the reaction in order to dissect heterochromatic and euchromatic tagmentations. However, these dual HP1-alpha-Tn5 activities proved to also be an advantage: the simultaneous detection of heterochromatin and euchromatin in the same cell made it possible to estimate the so-called “chromatin velocity”, a value in t-SNE graphs that shows the rate and direction of heterochromatin–euchromatin conversion. A big advantage of GET-seq is also the possibility to use it in a single cell approach and combine it with other methods such as scRNA-seq.

There is one intrinsic Tn5-assisted tagmentation weakness, arising from the fact that only 50% of the fragments obtained in the process are ready for PCR amplification, because the other 50% contain the same oligo on both sides, form a hairpin structure, and inhibit amplification. To increase the PCR yield, an elegant method was proposed [90]. Tn5 transposase was complexed with an adaptor containing uracile (U) immediately after the MEs that block polymerase to proceed beyond these points in the gap-filling reaction. In the second step, a locked dT-containing LNA primer was added, which is reverse-complement to the ME and contains a reverse adapter sequence 5′ overhang. The LNA primer has a higher melting temperature, preferably anneals to the mosaic end, and efficiently changes the i7 to the i5 primer sequence in the next PCR amplification step. Consequently, DNA fragments that initially contained two identical oligos after tagmentation are converted to fragments bearing two different sequencing primers.

In spatial transcriptomics, Tn5 tagmentation was used in the ATAC-see method, developed by the authors of the original ATAC-seq method itself, exploiting the idea of attaching a fluorescent oligo to the MEDS adapters [91]. This strategy resulted in effective tagmetation of open chromatin with the addition of fluorescent marks to them. These fluorescent loci could afterwards be seen in microscopy or FACS sorted for quantitative analysis of ATAC-seq.

## 6. Future Research Directions

As it was mentioned above, Tn5 transposase was very successfully used in omics methods. This is mainly due to its ability to form transpososome complexes in vitro, its extremely high efficiency of transposition of any pre-complexed adaptors, its unique enzymatic activity with zero turnover, and its lack of binding site bias during tagmentation. However, there are also weaknesses. For example, Tn5 is transposed in vivo into nucleosome-free regions of chromatin, i.e., mostly into euchromatin, while other transposases, such as Sleeping Beauty, can be inserted also into heterochromatin. This limitation can be circumvented, however, by constructing Tn5-containing hybrid molecules, as it was mentioned above.

Another direction is to create various hybrid molecules with different transposases, allowing them to be guided to specific regions of the genome for the tagmentation. Indeed, a number of interesting methods have already been developed. One of them uses a fusion protein between transcription factors and piggyBac transposase, allowing the transposition to occur in the vicinity of the binding sites of the corresponding transcription factor [92]. Another example placed a hybrid molecule between dCas9 protein and Sleeping Beauty SB100X (dCas9-SB100X) [93]. This allowed the redirection of the transposase to specific regions in the genome using single guide RNAs (sgRNA). In general, this is a very intensive area of research and a large number of new methods are constantly being developed.

## Figures and Tables

**Figure 1 mps-06-00024-f001:**
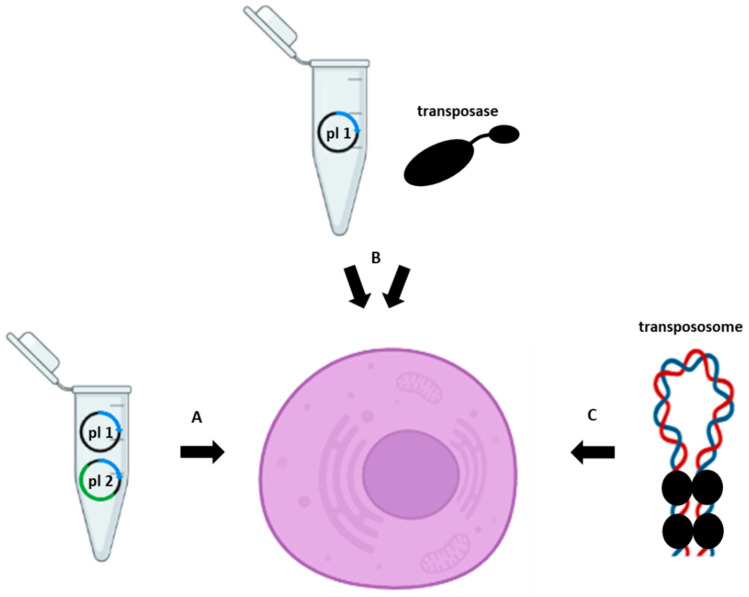
Transposase-assisted delivery of a gene of interest into the host cell using (**A**) two plasmids (plasmid one (pl 1) containing a transposon and plasmid two (pl 2) containing cDNA, encoding a transposase), (**B**) a plasmid (pl 1) containing a transposon and a transposase protein, or (**C**) a pre-assembled transpososome.

**Figure 2 mps-06-00024-f002:**
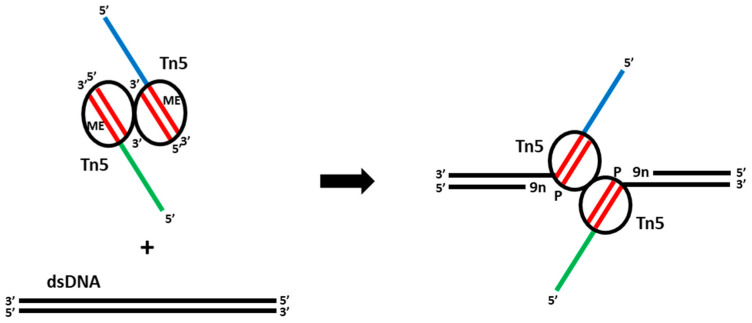
The mechanism of Tn5 tagmentation of DNA. Addition of a dimeric complex of Tn5 transposase loaded with MEDS adapters (mosaic ends (ME, red) and ssDNA overhangs (blue and green)) to dsDNA results in a DNA double-strand cut and MEDS attachment to the DNA leaving 9-nucleotide gaps (9n) that can be repaired in gap-filling reactions.

**Figure 3 mps-06-00024-f003:**
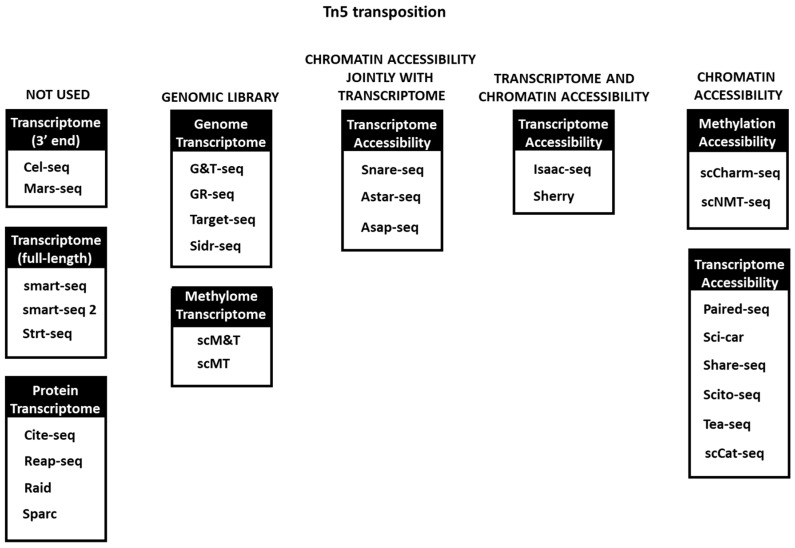
Tn5 transposition usage in multi-omics techniques. In some of the first omics methods it is not used or used only for DNA library preparation, while in others it is extensively used in one or several omics simultaneously.

**Figure 4 mps-06-00024-f004:**
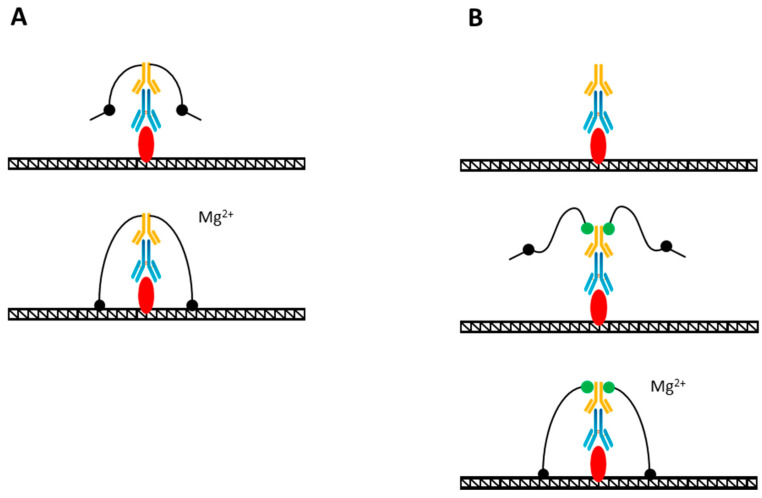
(**A**) TAM-ChIP method of genome profiling. The Tn5 adapter complex is pre-coupled to a secondary antibody (yellow) that is sequentially bound to a primary antibody (blue) that had been bound to the target (red). The addition of Mg ions (Mg^2+^) activates the tagmentation reaction. (**B**) CUT&Tag method. Tn5 adapter complex is covalently bound to protein-A (green) that is sequentially bound to a secondary antibody (yellow), primary antibody (blue), and the target (red). The tagmentation is activated by Mg ion (Mg^2+^) addition.

**Figure 5 mps-06-00024-f005:**
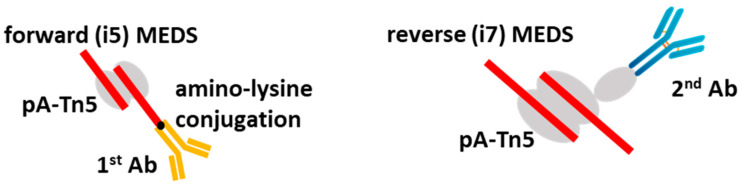
CUT&Tag method for multiple antibody usage in a single reaction (MulTI-Tag). Adapted from [85]. A primary antibody (yellow) is covalently attached to MEDS (red) containing the i5-compatible sequence MEDS, while a complex of a secondary antibody (blue) and protein-A-Tn5 contains an i7-compatible sequence MEDS.

## Data Availability

Data sharing not applicable.

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
