# Peer review of "Tn5 DNA Transposase in Multi-Omics Research"

_mps, 2023, doi:10.3390/mps6020024_

Round 1
Reviewer 1 Report
In principle, the paper is useful in its aim to review various Tn5 applications. However, the connection to other transposon systems, i.e. what has been done and can be done with other transposon systems, is inadequately and in some cases falsely covered. This is, in the first part of the paper, there exists a plethora of false statements, and I have tried to pinpoint them. Some key features of Tn5 usage are also missing, the most disturbing being the omission of gene delivery via electroporation of Tn5 transpososomes, and similarly, via electroporation of Mu transpososomes.
Astract:
Line 5. It stated already here in the abstract that Tn5 is UNIQUE in that it can form stable complexes with short adapters and transpose them without the aid of other proteins.
1. The uniqueness of Tn5 transposase is not true. In fact, the first DNA transposition system, in which this could be done is the DNA transposition system of phage Mu. In the seminal paper (Savilahti et al. 1995 EMBO Journal 14: 4893-4903) Kiyoshi Mizuuchi and his post-docs showed that two short DNA segments containing transposon end sequences only could be transposed into target DNA in vitro. This was catalyzed by MuA transposase only, and no other proteins were needed. The same strategy was somewehat later adapted for Tn5 transposition. This means that anything that can be done with Tn5 transposase can also be done with MuA transposase. MuA functions as a tetramer and the transpososome is more stable than Tn5 transpososome, which is dimeric. Especially for gene delivery into cells, this is beneficial. The seminal paper should be cited as the first transposition reaction made by the use of transposon end segments only; in vitro and solely with a single protein (MuA transposase).
2. The usage of ”adapters” is misleading as the active component of the system is composed of transposon end sequences into which any DNA can be attached. This is, the ”business” ends are in fact transposon ends that bind the transposase, and in Tn5 case form a dimeric synaptic compex, the Tn5 transpososome.
The authors should carefully study the literature for other systems, in which this can be done nowadays. I believe mariner system possibly is one, but there may be other similar systems as well.
Introduction.
Lines 17-24. It is true that all cut and paste transposons are of DDE-type. But it is not true other way round. For example, MuA transposase is of DDE-type (The Mu transpososome structure sheds light on DDE recombinase evolution: Montano et al. 2012 Nature 491, pp.413+), but it does not generate double-strand breaks (only the 3’-ends of the transposon are cleaved in the process called donor cleavage). When MuA is used in applications, this donor cleavage step is bypassed by the use of pre-cut transposon ends. In essense, the reaction series then mimicks cut and paste transposition.
Line 33. ”It is this feature (especially if their activity requires more than 2 different subunits) precludes many of them from a usage in vitro”
This is not true, Mu transpososome contains 4 MuA transposase subunits (Montano et al. 2012). There exists a vast literature for the usage of Mu in a plethora of in vitro transposition applications (see eg. Rasila et al. 2018 Nucleic Acids Res. 46: .4649-4661 and references therein). This paper and several others should be referenced (especially Haapa et al. papers 1999 Nucleic Acids Res and Genome Res)
Line 35. The list shoud definitely include also Mu transposition for a prokaryotic system. What is more, the Mu system can also be used in eukaryotic cells, namely in yeast and mammalian cells, including human ES cells (Paatero et al. 2008 Nucleic Acids Res. e148 ). Therefor Mu should be mentioned also in the context of eukaryotes.
Lines 36-38 ”Most of them, especially eukaryotic ones, in their native forms had no or very low activity, and many efforts have been done to revive them or to make them more active [5,6].”
MuA transposase, being a transposase of a transposing bacteriophage, is highly active already in its wild type form, as there is no biological selection pressure to reduce its activity. This is, the phage destroys its host anyway, and the transposon (disquised as a phage) escapes the cell. Yet, there was a room for improvement also in this system, and currently there are powerful hyperactice MuA variants available (Rasila et. al. 2017 Nucleic Acids Res. 9: 4649–4661), also commercially (Domus Biotechnologies, Turku, Finland; https://domusbiotechnologies.com/). The paper and commercial possibility should both be referenced.
Lines 39-43 ”The production of many active transposase forms made it possible to use 39 them in biotechnological applications, for example, as a tool to introduce various genes into a genome [7]. 40 This approach usually requires a two-vector system: one plasmid that encodes a transposon bearing a gene 41 to be transferred, and another plasmid that encodes the transposase under the control of a suitable 42 promoter (Figure 1). Alternatively, after delivery of transposon via a plasmid to a cell, the transposase can 43 be delivered as a protein.”
The description lacks the most powerful techniques to deliver genes via transposition, namely introduction of pre-assembled transpososomes. This can be done efficiently with Tn5 (for prokaryotic cells, Goryshin et al. 2000 Nature Biotechnology 18:97) and also with Mu (not only to gram-negative (Lamberg et al. 2002 Appl Env Microbiol 68:705-712) and gram-positive bacteria (Pajunen et al. 2005, Microbiology SGM 151: 1209-1218) but also to yeast and mammmalian cells Paatero et al. 2008 Nucleic Acids Res. 22: e148). The benefit with these techniques is that essentially all genomic integrations are single-copy. The above results should be added with the references given above).
Lines 53-56
Not only Tn5 is used for tagmentation. Mu-tagmentation is available commercially from Thermofisher.
Line 319. The term ”transposome complex” is used. The correct term is transpososome or DNA transposition complex.
Lines 322-324. ”However, other transposases have capabilities to transpose DNA into many other regions where Tn5 cannot; however, they have not been yet used exogenously. Therefore, one of the directions of research is to develop methods that will allow their usage for the delivery of pre-loaded DNA fragments to a given genomic regions.”
The logic of the sentences is odd, especially given the possibilities of in vitro integration and in vivo integration by many transposases. And e.g. Mu and Tn5 are capable for both of these tasks. The use of ”exogenous” is ambiguous and should de omitted. Use in every case ”in vitro” or ”in vivo”, as this terminology is unambiguous. The wording here requires substantial revision. Mu integration profile has been studied in vivo thorougly in yeast and mammalian cells (Paatero et al. 2008).
Author Response
Dear reviewer,
thank you very much for your thorough reading of the manuscript and your valuable comments.
Here there are my answers point-by-point.
- The uniqueness of Tn5 transposase is not true. In fact, the first DNA transposition system, in which this could be done is the DNA transposition system of phage Mu. In the seminal paper (Savilahti et al. 1995 EMBO Journal 14: 4893-4903) Kiyoshi Mizuuchi and his post-docs showed that two short DNA segments containing transposon end sequences only could be transposed into target DNA in vitro. This was catalyzed by MuA transposase only, and no other proteins were needed. The same strategy was somewehat later adapted for Tn5 transposition. This means that anything that can be done with Tn5 transposase can also be done with MuA transposase. MuA functions as a tetramer and the transpososome is more stable than Tn5 transpososome, which is dimeric. Especially for gene delivery into cells, this is beneficial. The seminal paper should be cited as the first transposition reaction made by the use of transposon end segments only; in vitro and solely with a single protein (MuA transposase). Response 1: I changed the corresponding statements, removed the word UNIQUE from the abstract, and added the information about MuA and the references (lines 79-83).
- The usage of ”adapters” is misleading as the active component of the system is composed of transposon end sequences into which any DNA can be attached. This is, the ”business” ends are in fact transposon ends that bind the transposase, and in Tn5 case form a dimeric synaptic compex, the Tn5 transpososome.Response 2: I removed the word "adapters" from the abstract and rephrased the sentences.
- Lines 17-24. It is true that all cut and paste transposons are of DDE-type. But it is not true other way round. For example, MuA transposase is of DDE-type (The Mu transpososome structure sheds light on DDE recombinase evolution: Montano et al. 2012 Nature 491, pp.413+), but it does not generate double-strand breaks (only the 3’-ends of the transposon are cleaved in the process called donor cleavage). When MuA is used in applications, this donor cleavage step is bypassed by the use of pre-cut transposon ends. In essense, the reaction series then mimicks cut and paste transposition.Response 3: I changed the text correspondingly adding the suggested references (lines 39-41).
- This is not true, Mu transpososome contains 4 MuA transposase subunits (Montano et al. 2012). There exists a vast literature for the usage of Mu in a plethora of in vitro transposition applications (see eg. Rasila et al. 2018 Nucleic Acids Res. 46: .4649-4661 and references therein). This paper and several others should be referenced (especially Haapa et al. papers 1999 Nucleic Acids Res and Genome Res) Response 4: I added this to the text with the suggested references (lines 81-82).
- The list shoud definitely include also Mu transposition for a prokaryotic system. What is more, the Mu system can also be used in eukaryotic cells, namely in yeast and mammalian cells, including human ES cells (Paatero et al. 2008 Nucleic Acids Res. e148 ). Therefor Mu should be mentioned also in the context of eukaryotes.Response 5: I included MuA in the list (line 52) and mentioned it in the context of eukaryotes.
- MuA transposase, being a transposase of a transposing bacteriophage, is highly active already in its wild type form, as there is no biological selection pressure to reduce its activity. This is, the phage destroys its host anyway, and the transposon (disquised as a phage) escapes the cell. Yet, there was a room for improvement also in this system, and currently there are powerful hyperactice MuA variants available (Rasila et. al. 2017 Nucleic Acids Res. 9: 4649–4661), also commercially (Domus Biotechnologies, Turku, Finland; https://domusbiotechnologies.com/). The paper and commercial possibility should both be referenced.Response 6: I changed the text correspondingly (lines 55-58) and added the reference of MuA commercial production (lines 88-89).
- The description lacks the most powerful techniques to deliver genes via transposition, namely introduction of pre-assembled transpososomes. This can be done efficiently with Tn5 (for prokaryotic cells, Goryshin et al. 2000 Nature Biotechnology 18:97) and also with Mu (not only to gram-negative (Lamberg et al. 2002 Appl Env Microbiol 68:705-712) and gram-positive bacteria (Pajunen et al. 2005, Microbiology SGM 151: 1209-1218) but also to yeast and mammmalian cells Paatero et al. 2008 Nucleic Acids Res. 22: e148). The benefit with these techniques is that essentially all genomic integrations are single-copy. The above results should be added with the references given above).Response 7: I added the mentioned results with the references (lines 67-69).
- Not only Tn5 is used for tagmentation. Mu-tagmentation is available commercially from Thermofisher. Response 8: I added this comment (line 88-89).
- The term ”transposome complex” is used. The correct term is transpososome or DNA transposition complex.Response 9: I changed it everywhere in the manuscript.
- The logic of the sentences is odd, especially given the possibilities of in vitro integration and in vivo integration by many transposases. And e.g. Mu and Tn5 are capable for both of these tasks. The use of ”exogenous” is ambiguous and should de omitted. Use in every case ”in vitro” or ”in vivo”, as this terminology is unambiguous. The wording here requires substantial revision. Mu integration profile has been studied in vivo thorougly in yeast and mammalian cells (Paatero et al. 2008).Response 10: I removed this sentence from the text of the manuscript.
Thank you very much!
Reviewer 2 Report
This review manuscript is well written and illustrates the focus of omics technology based on Tn5 Transposease. Recently, this field of research has been rapidly evolving by combining next-generation sequencing tools, epigenetic approaches, and single-cell-level analysis based on common ChIP-seq, RNA-seq, and genome-wide DNA-seq. This review deals with recent omics technology and introduces an applied analysis system. This type of review is appropriately needed to develop research interests and more ideal approaches. Basically, the Tn5 transposase works in the prokaryotic genome. Therefore, it may be necessary to add a subtitle and content for prokaryotic applications using Tn5 mutagenesis. This MS is suitable for publication after minor modifications.
- Added a section for prokaryotic approaches using Tn5 transposase-based omics technologies.
Minor comments
1. Already mentioned that need to add prokaryotic approaches of Tn5 mutagenesis and recent approaches.
2. Fig. 2 need modification for tagmentation scheme by pA-Tn5. Please add 3rd phases.
3. There are several miss-typos. Lines: 172, transposase; 312, ATAC-seq; reference #46 missed authors name.
4. Maybe needed to prepare abbreviation for short terms.
Author Response
Dear reviewer,
thank you very much for your thorough reading of the manuscript and your valuable comments.
Here there are my answers point-by-point.
- Already mentioned that need to add prokaryotic approaches of Tn5 mutagenesis and recent approaches.Response 1: I added some information about prokaryotic applications of Tn5 with the reference (lines 74-76).
- Fig. 2 need modification for tagmentation scheme by pA-Tn5. Please add 3rd phases.Response 2: This Fig. shows the principle of Tn5 tagmentation of DNA and to my opinion doesn't require the addition of a scheme of pA-Tn5 hybrid molecule usage. I show it later, in the Fig. 4.
- There are several miss-typos. Lines: 172, transposase; 312, ATAC-seq; reference #46 missed authors name.Response 3: I changed it in the text. In the line 312 the correct name is ATAC-see (with the reference describing the method)
- Maybe needed to prepare abbreviation for short terms.Response 4: I can include them if there is a necessity from the journal.
Thank you very much!
Reviewer 3 Report
In the review manuscript entitled “Tn5 DNA transposase in multi-omics research” the authors reviewed the utility, advances in application in multi-omics technologies. Below are the comments.
1. The authors should provide a little background on transposons (definition, history, significance), and the transitioning to transposases can be done.
2. Lines# 321-322: “However, other transposases have capabilities to transpose DNA into many other regions where Tn5 cannot; however, they have not been yet used exogenously.” What is the exact meaning of this statement? The authors should explain.
3. The authors should also include the disadvantages of Tn5 DNA transposases and how they can be circumvented to render them more versatile.
Author Response
Dear reviewer,
thank you very much for your thorough reading of the manuscript and your valuable comments.
Here there are my answers point-by-point.
- The authors should provide a little background on transposons (definition, history, significance), and the transitioning to transposases can be done.
Response 1: I added the background in the first paragraph of the text (lines 17-31).
2. Lines# 321-322: “However, other transposases have capabilities to transpose DNA into many other regions where Tn5 cannot; however, they have not been yet used exogenously.” What is the exact meaning of this statement? The authors should explain.
Response 2: I removed this sentence from the text of the manuscript.
3. The authors should also include the disadvantages of Tn5 DNA transposases and how they can be circumvented to render them more versatile.
Response 3: I added this comment (lines 354-357).
Thank you very much!
Round 2
Reviewer 1 Report
Dear authors. The paper is now in much better shape. However, certain details were still wrong in the common transposon introduction part of the paper. I have tried to edit the text to be accurate and factful in its content. I hope you agree with the edits.
The Tn5 usage part is better and I do not have many points there.
As the suggested corrections are in some cases relatively substantial, I'll include a PDF file with text in colored font: Red font- delete. Blue font- suggested text. Green font: comments. I hope these are helpful.
A few important points:
As the transpososome delivery into cells is now described thoroughly, it should be added in the figure as well.
Check carefully all the references, so that the referring is correct and accurate. I added a couple of them. Mizuuchi 1983 is highly relevant first in vitro paper. McClintock made the discover already during 1940's, and there is a suggested reference for that, too

Author Response
Dear reviewer,
thank you very much for your help to make the manuscript better.
I changed the text accordingly.
I also changed the Fig 1
I updated the references.
Thanks again!
Dmitry
Round 3
Reviewer 1 Report
Dear authors. The paper is almost ready. However, some misuderstandings were still present in the manuscript, and I corrected them. Some references were missing and some were inaccurate, I also corrected them and also suggested some relevant ones. As earlier, my suggestions are in the attached file. Blue font: suggested text, Red font: delete, Green font: comments

Author Response
Dear reviewer,
Thank you very much for the comments.
I changed the text accordingly.
Round 4
Reviewer 1 Report
The paper is now ready for publication with the minor correction (below).
One minor thing. The figure legend still has "transpososome complex", which is kind of a redundant phrase. Preferentially use "pre-assembled transpososome" or "pre-assembled DNA transposition complex".
Author Response
Dear reviewer,
thank you very much!
I changed the Figure legend as you suggested.